# A Spatio-Temporal Flow Model of Urban Dockless Shared Bikes Based on Points of Interest Clustering

**Jian Dong [1] , Bin Chen [1] , Lingnan He [2,3,*], Chuan Ai [1] , Fang Zhang [1] , Danhuai Guo [4] and Xiaogang Qiu [1]**

[1]  College of Systems Engineering, National University of Defense Technology, Changsha 410073, China
[2]  The School of Communication and Design, Sun yat-sen University, Guangzhou 510000, China
[3]  Guangdong Key Laboratory for Big Data Analysis and Simulation of Public Opinion, Guangzhou 510006, China
[4]  Scientific Data Center, Computer Network Information Center, Chinese Academy of Sciences,
    4th South Fourth Road Zhongguancun, Beijing 100190, China
**\***  Correspondence: heln3@mail.sysu.edu.cn

**Abstract:** With the advantages of convenient access and free parking, urban dockless shared bikes are favored by the public. However, the irregular flow of dockless shared bikes poses a challenge for the research of flow pattern. In this paper, the flow characteristics of dockless shared bikes are expounded through the analysis of the time series location data of ofo and mobike shared bikes in Beijing. Based on the analysis, a model called *DestiFlow* is proposed to describe the spatio-temporal flow of urban dockless shared bikes based on points of interest (POIs) clustering. The results show that the *DestiFlow* model can find the aggregation areas of dockless shared bikes and describe the structural characteristics of the flow network. Our model can not only predict the demand for dockless shared bikes, but also help to grasp the mobility characteristics of citizens and improve the urban traffic management system.

**Keywords:** dockless shared bike; points of interest; aggregation area; spatio-temporal flow model

## 1. Introduction

Shared bikes are the products of the shared economy and the development of the Internet of things. As a supplement to the urban traffic network, shared bikes not only solve the problem of "the first/last kilometer" for citizens to travel but also effectively alleviate traffic congestion and environmental pollution [1,2]. The development of shared bikes has experienced a process from docked shared bikes to dockless shared bikes. The docked shared bike is a means of transportation, which can only be rented and returned at certain fixed docked stations. Citizens need to concern about the limitation of docks when they want to use or return the bikes. Different from the docked shared bikes, the dockless shared bikes, such as ofo and mobike, have developed rapidly in China because the advantages of convenient access and free parking. Since the dockless shared bikes can be accessed and parked at any valid place, their trajectories can more truly reflect the behaviors of citizens on short trips. However, it becomes more difficult to study the flow rules of the dockless shared bikes. More accurate description and advance prediction of the flow can not only predict the demand for dockless shared bikes, but also help to grasp the mobility characteristics of citizens and improve the urban traffic management system.

Most of the existing works focus on the research of docked shared bikes, which mainly include the distribution rebalancing [3–6], station optimization [7,8], demand prediction [9–12], and flow prediction [13]. However, there are only few studies on dockless shared bikes. Pan et al. used the deep reinforcement learning framework to motivate users materially and achieve the goal of rebalancing the distribution of dockless shared bikes [14]. Liu et al. used the method of deep learning to infer

the distribution of dockless shared bikes in a new city [15]. These studies are based on the regular geographic grid, and the dockless shared bikes are assigned to corresponding grids according to their locations. Although this method is easy to operate, there are the following problems: the method using regular geographic grid damages the integrity of geographical space and splits the clustered shared bikes; the flow of dockless shared bikes are not constrained by the regular geographic grid; the behavior of citizens is complex; and the regular geographic grid greatly simplifies this complex behavior. Therefore, the method using regular geographic grid may affect the accuracy of data analysis.

The flow model of shared bikes is mainly used to describe the flow rules of shared bikes in time and space. The flow characteristics of shared bikes are easy to be affected by the climate, terrain, infrastructure, and other objective factors. Meanwhile, the subjective factors of citizens, such as safety perceptions, convenience, time valuation, exercise valuation and habits, will also affect the demand for shared bikes [16,17]. These objective and subjective factors add difficulty to the construction of flow model of shared bikes. However, there are some interesting rules found by the studies of shared bikes. In the short term, the flow of shared bikes is periodic [10,18], and the weekly flow pattern is similar. In addition to obvious differences between weekdays and weekends, the daily flow pattern is similar on the weekdays or weekends. Distance is an important factor that limits the usage and destination selection of shared bikes. Therefore, many destination selection models [19,20] take distance as the first-choice factor. The existing flow models [21,22] focus on the abstract patterns of human mobility behavior. Although the flow of shared bikes is created by human mobility, it is a special mobility behavior. As a result, these models are not suitable for describing the flow of shared bikes with specific characteristics. Although Hoang et al. combined the mobile data of dockless shared bikes with weather data and network data to study the citywide crowd flow of docked shared bikes by dividing the city into several regions [13]. This method describes the flow of shared bikes from a macroscopic view, and it cannot describe the flow situation of shared bikes in detail.

In this paper, a novel method is presented to describe the flow of the urban dockless shared bikes. The dockless shared bikes have no station restrictions, making it difficult to characterize their flow. However, the spatial distribution of dockless shared bikes presents aggregation effect, and the aggregation areas can be found to study the flow of dockless shared bikes. In this paper, the flow characteristics of dockless shared bikes are expounded through the analysis of the time series location data of ofo and mobike shared bikes in Beijing. Based on the analysis, a model called *DestiFlow* is proposed to describe the spatio-temporal flow of urban dockless shared bikes. The model can be divided into three parts: the POI-based clustering, spatial flow distribution model, and time distribution model. The POI-based clustering is used to find the aggregation areas of dockless shared bikes, this method avoids the problems of regular geographic grid method effectively. On the basis of the aggregation areas, the spatial distribution model of dockless shared bikes is constructed according to the characteristics of flow distance and the activity of each aggregation area. The spatial flow distribution model determines the departure and arrival aggregation areas of each flow. According to the records of the flow of dockless shared bikes, the probability model is used to describe the time distribution of flow. The time distribution model schedules the number of dockless shared bikes within a certain period.

The rest of this paper is organized as follows: The second section introduces the data set and analyzes the flow characteristics of dockless shared bikes. The third section proposes the spatio-temporal flow model of urban dockless shared bikes based on POIs clustering called *DestiFlow*. The fourth section evaluates the POI-based clustering and the *DestiFlow* method. The fifth section carries on a case analysis and the sixth section gives the summarization of the paper.

## 2. Data Description and Analysis

### 2.1. Dataset Description

We collect the time series location data of ofo and mobike shared bikes in Beijing (longitude range [116.178603739, 116.56741803], latitude range [39.756344012, 40.034994274]) through the application program interface (API) of WeChat applet. The time span of ofo dataset is six days from 5 December 2017 to 10 December 2017, containing 16 million records and five hundred thousand ofo shared bikes. The weather of Beijing in these six days were clear and cloudy with an average maximum temperature of 6.0 °C and an average minimum temperature of −4.3 °C. The time span of mobike dataset is six days from 22 May 2018 to 27 May 2018, containing 20 million records and four hundred thousand mobike shared bikes. The weather of Beijing in these six days were clear and cloudy with an average maximum temperature of 29.7 °C and an average minimum temperature of 17.0 °C. The ofo dataset is denoted as $D_1$, and the mobike dataset is denoted as $D_2$. The two datasets have the same structure, including the unique identification of the dockless shared bikes, the latitude and longitude, and the time in that latitude and longitude.

The locations of a dockless shared bike change at different times, and every two adjacent locations are considered to constitute a flow. The time series location data of dockless shared bikes can be affected by various external factors, such as GPS drift, truck haulage, etc., all of which will cause the location errors. To eliminate the influence of these factors on data quality, the flow distance, flow time and average flow speed are constrained according to the flow characteristics of dockless shared bikes. The range of flow distance, flow time and average flow speed is [0.4 km, 20 km], [2 min, ∞], and [5 km/h, 25 km/h], respectively. Since what we have obtained are the static positions of dockless shared bikes, the actual routes created by the bikes are unknown. This paper uses Manhattan distance as the actual distance. In urban traffic, the Manhattan distance between two points is close to their real distance [23].

### 2.2. Flow Characteristics of Dockless Shared Bikes

The distance characteristics of the dockless shared bikes are analyzed firstly. The development of dockless shared bikes solves the problem of "the first/last kilometer" for citizens to travel. Therefore, the flow distance of dockless shared bikes is different from other transportation. Figure 1 shows the distribution of the flow distance. The black line is the fitting line of the distance distribution with the lognormal distribution, and the goodness of fit for $D_1$ and $D_2$ is 0.924 and 0.931, respectively. The flow distance distribution of dockless shared bikes can be described by a lognormal distribution. Figure 1 indicates that the dockless shared bike is a short-distance transportation. Its travel distance is mainly within 5 km, and about 1 km is the high-frequency distance. The average flow distance in winter is obviously lower than that in summer, which is consistent with the results of Reference [24]. The flow distance distribution of docked shared bikes, such as France Vélo'v [18], is segmented, because the distance has a close relationship with the charge. However, dockless shared bikes do not show this characteristic obviously.

To further analyze the time characteristics of flow distance, the flow distance distributions in different time windows are compared using JS divergence [25]. JS divergence is an improvement of KL divergence [26] and makes up for the asymmetry of KL divergence. JS divergence is an indicator to measure the difference between two distributions. The smaller the JS divergence is, the smaller the difference between the two distributions will be. JS divergence is defined as follows:

$$JS(D^t||D^{t+1}) = \frac{1}{2}KL(D^t||\frac{D^t + D^{t+1}}{2}) + \frac{1}{2}KL(D^{t+1}||\frac{D^t + D^{t+1}}{2}). \tag{1}$$

where,

$$KL(D^t||D^{t+1}) = \sum_{i \in I} D^t(i)log\frac{D^t(i)}{D^{t+1}(i)}, \tag{2}$$

$D^t$ is the flow distance distribution of dockless shared bikes in the time window $[t, t+1)$. The flow distance is divided into 49 intervals, each of which is 400 m in length, $I$ = [400 m, 800 m), [800 m, 1200 m), ..., [19,600 m, 20,000 m]. $D^t(i)$ represents the proportion of the distance within the range $i \in I$ in the time window $[t, t+1)$.

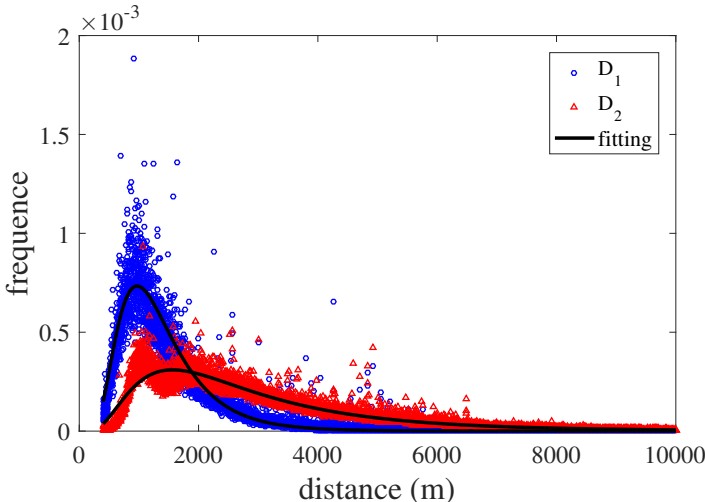

**Figure 1.** The distribution of the flow distance of dockless shared bikes. The blue circle represents the distance distribution obtained from $D_1$, and the red triangle represents the distance distribution obtained from $D_2$. The black line is the fitting line of the distance distribution with the lognormal distribution. The flow distance obtained from $D_1$ obeys a lognormal distribution with the parameters $\mu = 7.123$, $\sigma = 0.496$, and the flow distance obtained from $D_2$ obeys a lognormal distribution with the parameters $\mu = 7.793$, $\sigma = 0.662$.

Figure 2 shows the JS divergence of flow distance distribution at different times, the time window of the blue line is one hour, and the time window of the red line is 12 h. When the time window is one hour, the average JS divergence calculated from $D_1$ and $D_2$ is 0.0039 and 0.0093, respectively. When the time window is 12 h, the average JS divergence calculated from $D_1$ and $D_2$ is 0.0023 and 0.0009, respectively. Although the values of JS divergence fluctuates with time, they are very small, and the maximum value is 0.0478. The larger the time window is, the smaller the difference of flow distance distribution is. Figure 2c shows the two distance distributions when JS divergence is the maximum, indicating the flow distance distribution of dockless shared bikes does not change obviously with time, it is time stable.

Figure 3 shows the change in the number of departures of dockless shared bikes. As can be seen from the figure, the number of departures varies periodically, and there is a significant difference between weekdays and weekends. There are two obvious peaks in daily departures, which are around 8:00 and 18:00, respectively, indicating that these two periods are the rush hours for citizens to travel. The number of departures declines rapidly after the rush hours. Compared with changes in the number of departures obtained from $D_1$ and $D_2$, $D_2$ has more obvious peaks at about 8:00 and 18:00, but its values from 10:00 to 16:00 are lower than $D_1$. The reason is that the temperature around 8:00 and 18:00 in summer is comfortable and conducive to travel, while the temperature at noon in winter is more comfortable. Therefore, the temperature has a great influence on the demand for dockless shared bikes, and citizens tend to use the bikes when the temperature is comfortable.

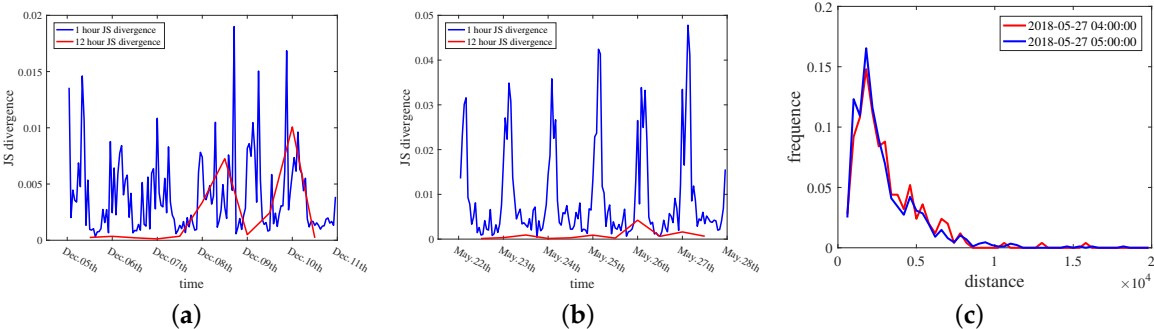

**Figure 2.** JS divergence of flow distance distribution at different times and the comparison between two distance distributions when JS divergence is the maximum. (**a**) JS divergence calculated from $D_1$. (**b**) JS divergence calculated from $D_2$. The time window of the blue line is 1 h, and the time window of the red line is 12 h. Each data point is the JS divergence calculated by the flow distance distribution in the current time window and that in the previous time window. (**c**) The comparison between two distance distributions when JS divergence is the maximum. The times of the two distance distributions are 04:00:00 and 05:00:00 on 27 May 2018 respectively.

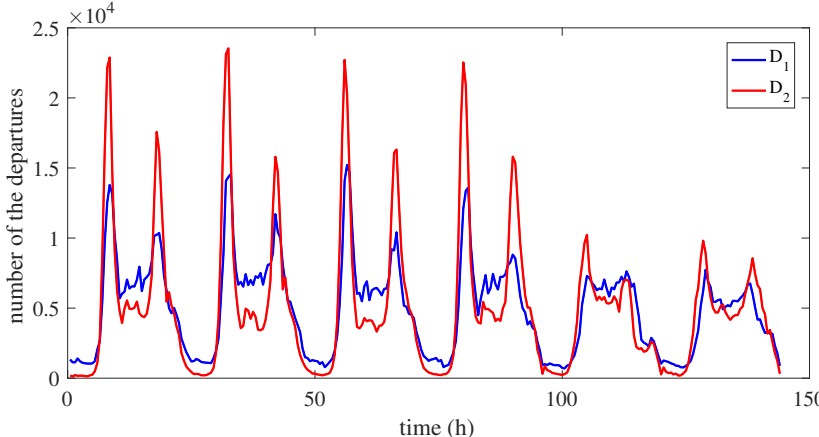

**Figure 3.** The change in the number of departures of dockless shared bikes. The horizontal axis represents the time interval between the current time in the dataset and the start time. The ordinate axis represents the number of departures within 30 min. The blue line represents the change in the number of departures obtained from $D_1$, and the red line represents the changes in the number of departures obtained from $D_2$.

Through the above analysis, we can conclude: (1) The flow distance distribution of dockless shared bikes follows a lognormal distribution, and the flow distance distribution is time stable; (2) the flow characteristics of weekdays and weekends are quite different. But in the short term, the flow is periodic, and there are two peak periods every day, around 8:00 and 18:00; (3) the temperature has a great influence on the demand for dockless shared bikes, and citizens tend to use the bikes when the temperature is comfortable.

## 3. Destiflow

In this section, we construct a spatio-temporal flow model of urban dockless shared bikes based on POIs clustering called *DestiFlow*. The model can be divided into three parts: the POI-based clustering, spatial flow distribution model, and time distribution model. The POI-based clustering is used to find the aggregation areas of dockless shared bikes, the spatial flow distribution model determines the departure and arrival aggregation areas of each flow, and the time distribution model schedules the number of bikes within a certain period.

The time series location data of dockless shared bikes can be expressed as:

$$X = (bid, lng_B, lat_B, \tau). \tag{3}$$

where $bid$ represents the unique identification of dockless shared bikes, $lng_B$ represents the longitude, $lat_B$ represents the latitude, and $\tau$ represents the time in that latitude and longitude. The aggregation areas $A(X)$ of dockless shared bikes can be extracted based on the POIs clustering. The aggregation area set of $X$ can be expressed as:

$$A(X) = (aid, lng_A, lat_A, \tau). \tag{4}$$

where $aid$ represents the unique identification of aggregation areas, $lng_A$ represents the central longitude, $lat_A$ represents the central latitude.

A single flow of dockless shared bikes can be represented by an aggregation area pair:

$$f = (A_d, A_a). \tag{5}$$

where $A_d$ represents the aggregation areas where dockless shared bikes depart and $A_a$ represents the aggregation areas where dockless shared bikes arrive, $\tau_{A_d} < \tau_{A_a}$. In a certain time interval $T$, the spatio-temporal flow of dockless shared bikes can be described as:

$$flow_{ts} = \{f_1, f_2, \ldots, f_C\}. \tag{6}$$

$$\tau_{f_1 \cdot A_d} \leqslant \tau_{f_2 \cdot A_d} \leqslant \ldots \leqslant \tau_{f_C \cdot A_d}. \tag{7}$$

where $C$ represents the total number of flow, and formula 7 ensures the time-ordered of the flow.

Suppose that departure aggregation areas $A_d$ and arrival aggregation areas $A_a$ follow two different distributions:

$$A_d \sim Dep : P(A_d = i). \tag{8}$$

$$A_a \sim Arr : P(A_a = j | A_d = i). \tag{9}$$

where $Dep$ and $Arr$ respectively represent two different distribution functions, $i$ and $j$ represent the index of aggregation areas, $i \neq j$. Then the spatial flow distribution of dockless shared bikes can be expressed as:

$$flow_s \sim Dep \cdot Arr. \tag{10}$$

The flow of dockless shared bikes is sequential, and a time window $[t, t + \Delta t)$ is set. By constructing the time distribution of flow, the spatial flow distribution of each time window $flow_s^{\Delta t}$ can be established:

$$flow_s^{\Delta t} = (A_d, A_a) | (\tau_{A_d}, \tau_{A_a} \in [t, t + \Delta t)) \wedge (A_d, A_a \in flow_s). \tag{11}$$

All $flow_s^{\Delta t}$ in time interval $T$ constitute the spatio-temporal flow $flow_{ts}$. After that, the POI-based clustering, spatial flow distribution model and the time distribution model are introduced in detail.

## 3.1. POI-Based Clustering

Although the dockless shared bikes have no station restrictions, their spatial distribution presents aggregation effect. Therefore, the aggregation areas can be found to study the flow of dockless shared bikes between aggregation areas. In the geographic information system, a POI is a specific location that someone finds useful or interesting. A POI can be a house, a shopping mall, a subway station, etc., and the aggregation areas of dockless shared bikes are also distributed around these locations. A POI-based clustering method is proposed to find the aggregation areas.

In a certain area $\Gamma$, the location sample data is expressed as $G = \{g_1, g_2, \ldots, g_K\}$, and $K$ is the number of samples. The set of POIs is expressed as $P = \{p_1, p_2, \ldots, p_H\}$, and $H$ is the number of POIs. The POI-based clustering is described as follows:

**Step1**: For each sample data $g_i$, find the closest POIs:

$$p(g_i) = \arg \min_{p_j \in P}(d_{g_i, p_j}). \tag{12}$$

where $d_{g_i, p_j}$ represents the distance between $g_i$ and $p_j$. Each sample can be assigned to the POIs closest to it.

However, many POIs are very close to each other, so the region of the sample cannot be determined only by the distance between the sample location and the POIs. On the basis of the above, the POIs that have been assigned to the sample are clustered, and the POIs close to each other are regarded as the same cluster.

**Step2**: Use k-means clustering method to get the class of each POI $A(p_i)$;

**Step3**: Map the sample data to the class of POIs $A(G)$.

For the time series location data of dockless shared bikes $X$, the clustering regions obtained by the above method are regarded as aggregation areas of dockless shared bikes $A(X)$.

## 3.2. Spatial Flow Distribution Model

There are two types of behaviors of dockless shared bikes, departure and arrival. Activity index ($AI$) is defined based on the number of departures and arrivals. The activity of an aggregation area is expressed as the sum of the number of departures and arrivals in the aggregation area in a time interval:

$$AI_A^{\Delta t} = dep_A^{\Delta t} + arr_A^{\Delta t}. \tag{13}$$

where $dep_A^{\Delta t}$ and $arr_A^{\Delta t}$ represent the number of departures and arrivals of aggregation area $A$ in the time interval $\Delta t$, respectively. The activity index is the representation of the spatial characteristics, which reflects the travel choice of citizens. The more active an aggregation area is, the more popular the aggregation area is, and the greater the probability of departure and arrival from there is. In addition, distance is an important factor affecting the usage of dockless shared bikes and the distribution of arrival aggregation areas.

Therefore, we assume that the distribution of departure aggregation areas and arrival aggregation areas of dockless shared bikes can be constructed with the factors activity and distance:

$$Dep : P(A_d = i) = \frac{(AI_i)^\alpha}{\sum_j (AI_j)^\alpha}. \tag{14}$$

$$Arr : P(A_a = j | A_d = i) = \Psi(d_{i,j}) \cdot \frac{(AI_j)^\beta}{\sum_k (AI_k)^\beta}. \tag{15}$$

where $\alpha$ and $\beta$ are power exponents, $d_{i,j}$ represents the Manhattan distance between aggregation area $i$ and aggregation area $j$, and $\Psi(d_{i,j})$ is a function of $d_{i,j}$. Based on the analysis of the flow distance of dockless shared bikes, the distance distribution is time-stable, and can be described by a lognormal distribution. Therefore, we assume that:

$$\Psi(d_{i,j}) = \frac{1}{\sqrt{2\pi}\sigma d_{i,j}} exp[-\frac{(ln(d_{i,j}) - \mu)^2}{2\sigma^2}]. \tag{16}$$

where $\mu$ and $\sigma$ are two parameters, which can be obtained through training.

### 3.3. Time Distribution Model

The total number of flow $C$ can be obtained through the number of dockless shared bikes $\Omega$ and the usage rate $\lambda$:

$$C = \lambda \cdot \Omega. \tag{17}$$

We assume that the number of dockless shared bikes is constant for a long time. Figure 3 indicates that the usage of dockless shared bikes is nearly constant in a short time (weekdays or weekends).

The number of flow in the time window $[t, t + \Delta t)$ is:

$$|flow_s^{\Delta t}| = C \cdot \Phi(t, t + \Delta t). \tag{18}$$

where $\Phi(t, t + \Delta t)$ represents the flow probability in time window $[t, t + \Delta t)$.

Based on the observation of Figure 3, the flow probability is different at each time, and there are two peaks in a day. A Gaussian mixture model is used to model the flow time distribution of dockless shared bikes in a day. The Gaussian mixture model is a statistical model that quantifies the distribution of variables accurately with multiple gaussian probability density functions. It is usually used to approximate the data distribution [27–29]. The one-dimensional Gaussian mixture model can be expressed as:

$$\varphi(t) = \sum_{i=1}^{K} \theta_i \frac{1}{\sqrt{2\pi}\sigma_i} exp[-\frac{(t - \mu_i)^2}{2\sigma_i^2}]. \tag{19}$$

where $\theta_i$ represents the coefficient of each Gaussian distribution component, and $\sum_{i=1}^{K} \theta_i = 1$. $\mu_i$ is the mean of the component $i$, $\sigma_i$ is the standard deviation of the component $i$. Therefore, the flow probability in the time window $[t, t + \Delta t)$ can be expressed as:

$$\Phi(t, t + \Delta t) = \int_{t}^{t+\Delta t} \varphi(x)dx. \tag{20}$$

## 4. Model Evaluation

### 4.1. Evaluation of Clustering Model

To illustrate the effect of the POI-based clustering, we compare it with the method based on regular geographic grid. We take the time series location data of ofo shared bikes on 5 December 2017 (a total of 437,059 locations) as sample data. The two methods are used to cluster the sample data separately. The number of clusters are set to 100, 500, 1000, 5000, 10,000, 20,000. The clustering verification methods are used to evaluate the clustering results of aggregation areas. The commonly used clustering verification methods are Calinski–Harabasz index (CH) [30], Davies–Bouldin index (DB) [31] and I index (I) [32], which are defined as:

Calinski–Harabasz index:

$$CH = \frac{\sum_i n_i d^2 (c_i, c) / (NC - 1)}{\sum_i \sum_{x \in C_i} d^2 (x, c_i) / (n - NC)}. \tag{21}$$

Davies–Bouldin index:

$$DB = \frac{1}{NC} \sum_i \max_{j, j \neq i} \left\{ \left[ \frac{1}{n_i} \sum_{x \in C_i} d(x, c_i) + \frac{1}{n_j} \sum_{x \in C_j} d(x, c_j) \right] / d(c_i, c_j) \right\}. \tag{22}$$

I index:

$$I = \left( \frac{1}{NC} \cdot \frac{\sum_{x \in D} d(x, c)}{\sum_i \sum_{x \in C_i} d(x, c_i)} \cdot \max_{i,j} d(c_i, c_j) \right)^P. \tag{23}$$

where $D$ represents sample data, $c$ represents the center of sample data, $P$ represents the dimension of sample data, $NC$ represents the number of clusters, $C_i$ represents the ith cluster, $n_i$ represents the

number of objects in $C_i$, $c_i$ represents the center of $C_i$, and $d(x, y)$ represents the distance between $x$ and $y$. These three methods reflect the degree of tightness within and between clusters from different perspectives. The higher the $CH$ and $I$ are, the better the clustering effect will be. The smaller the DB is, the better the clustering effect will be.

Figure 4 shows the comparison of the POI-based clustering and regular geographic grid. Figure 4a shows the result that the sample data is divided into 100 clusters using the method using regular geographic grid, and Figure 4b shows the result that the sample data is divided into 100 clusters using the POI-based clustering method. It is found that the POI-based clustering is more flexible and reasonable in the division of geographic space. Figure 4c–e show the evaluation of the POI-based clustering and regular geographic grid by CH, DB, and I, respectively. The results show that the POI-based clustering is better to the method based on geographic grid, which suggests that the POI-based clustering is more effective to find the aggregation areas of dockless shared bikes.

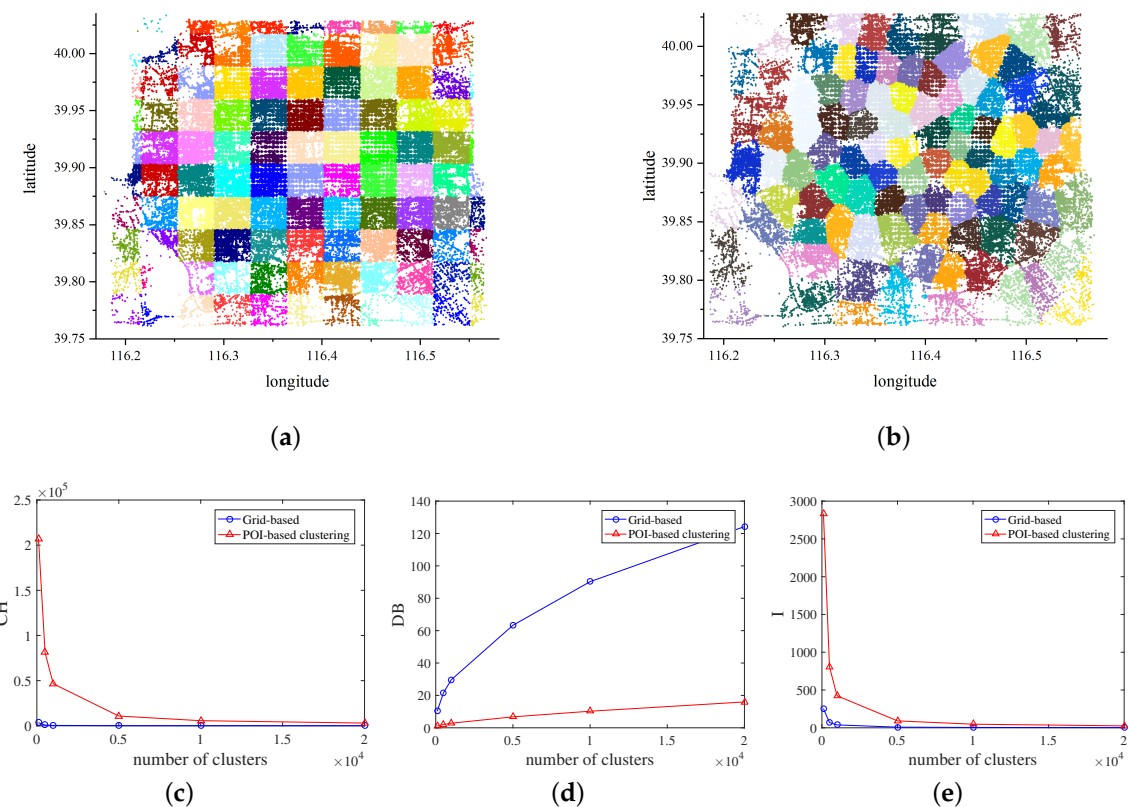

**Figure 4.** The comparison of the points of interest (POI)-based clustering and regular geographic grid. (**a**) The sample data is divided into 100 clusters using the method using regular geographic grid. The horizontal axis represents longitude and the ordinate axis represents latitude. Different colors in the figure represent different clusters. (**b**) The sample data are divided into 100 clusters using the POI-based clustering method. (**c**) The Calinski–Harabasz index at different clusters. (**d**) The Davies–Bouldin index at different clusters. (**e**) The I index at different clusters.

## 4.2. Evaluation of DestiFlow

According to the analysis in the second section, the flow characteristics of dockless shared bikes on the weekdays are different from those on the weekends, so the weekdays and weekends are considered separately. In this section, the empirical data on 5 December (Tuesday) and 10 December (Sunday) are selected to evaluate the spatio-temporal flow model of urban dockless shared bikes based on POIs clustering.

### 4.2.1. Setup

The model parameters are obtained through empirical data. The input of the model includes the spatial distribution of the aggregation areas, the activity of the aggregation areas, the time distribution parameters and the distance distribution parameters. Although Figure 4 shows that the smaller the number of clusters, the better the clustering effect, there will be more dockless shared bikes flowing in the same aggregation areas. In order to make the flow of dockless shared bikes more in different aggregation areas, the number of aggregation areas is set to 10,000, so that only 0.1% of the flow is in the same aggregation area. The activity of each aggregation area can be obtained from the data. Figure 5 shows the fitting results of the time distribution on 5 December and 10 December respectively using the one-dimensional Gaussian mixture model. The model parameters are shown in the upper left corner of each subfigure. The lognormal distribution is used to fit the distance distribution on 5 December and 10 December (the fitting result is similar to that in Figure 1), and the distance distribution parameters could be obtained. The distance distribution on 5 December and 10 December is subject to $lognormal(7.053, 0.496)$ and lognormal $(7.303, 0.424)$ respectively.

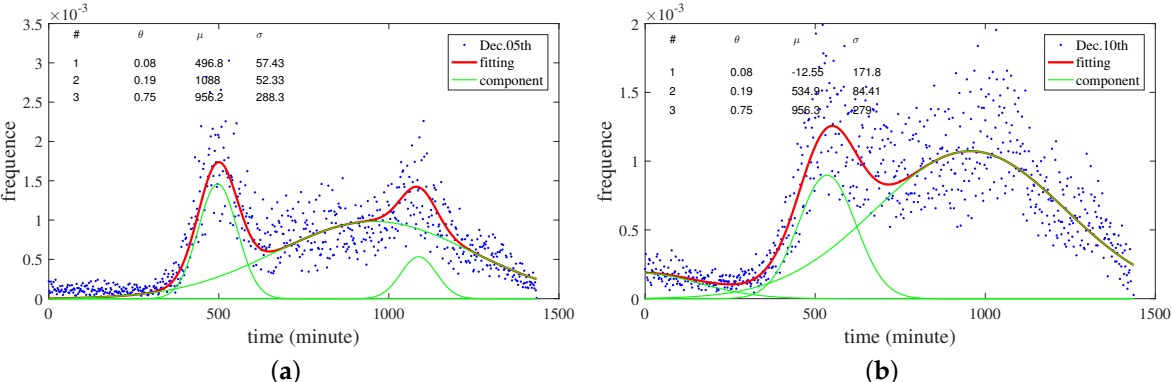

**Figure 5.** The time distribution of flow on 5 December and 10 December. (**a**) time distribution of flow on 5 December (**b**) time distribution of flow on 10 December. The blue dots in the figure represent the proportion of flow per minute, and the time distribution is fitted by the 3rd-order Gaussian mixture model in the one-dimensional space. The goodness of fit for December 05 is 0.7687, and the goodness of fit for 10 December is 0.7684. The green lines represent components of the Gaussian mixture model. The model parameters are shown in the upper left corner of each subfigure.

The $\alpha$ and $\beta$ are variable parameters. It is necessary to select the best parameters for subsequent experiments. The maximum likelihood approach is used to select the optimal parameters of the model. The maximum likelihood approach is usually used to compare a series of models numerically and select the best parameters to interpret the data. It has been widely used in estimating model parameters [33–35].

Estimating the likelihood of the *DestiFlow* method in this paper involves considering the possibility $P(f_i)$ of each flow $f_i$ in real data according to the model. The likelihood function can be expressed as:

$$P(f) = \prod_i P(f_i). \tag{24}$$

To obtain better numerical accuracy, log-likelihood is used in this paper:

$$\log\left(\prod_i P(f_i)\right) = \sum_i \log(P(f_i)). \tag{25}$$

Figure 6 shows the relationship between the log-likelihood of *DestiFlow* and parameters $\alpha$ and $\beta$. The log-likelihoods of the *DestiFlow* method are a convex function of model parameters $\alpha$ and

$\beta$, so the maximum likelihood can be found to estimate the best parameters of the model. It can be found from the figure that, for weekdays and weekends, the likelihood is the maximum when $\alpha = 1.0$ and $\beta = 1.0$. Therefore, this pair of parameters is selected for subsequent experiments. The model parameters are loaded into the model for the experiments. To simulate the activity of shared bikes in one day, the experiments start at 00:00, end at 24:00, and the time window is 30 min.

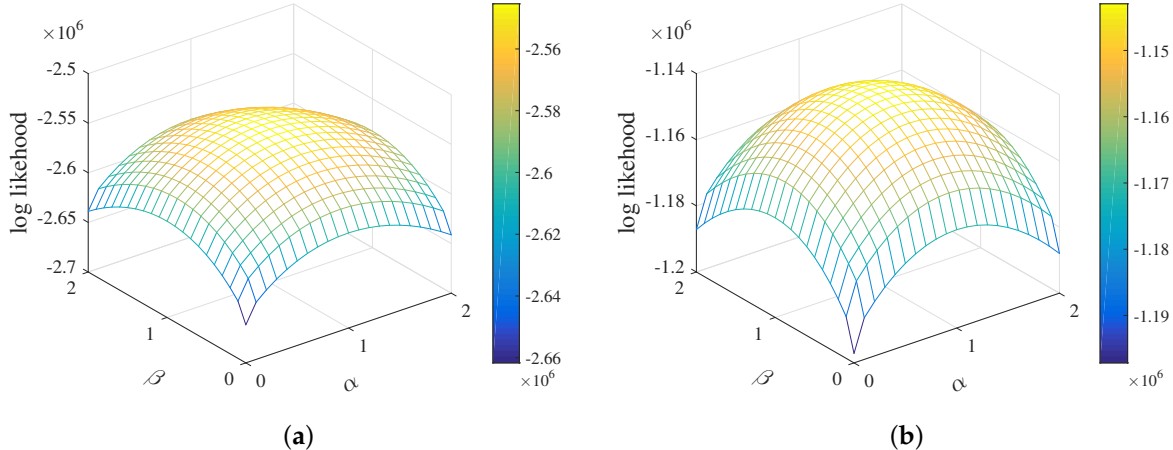

(**a**)  (**b**)

**Figure 6.** The relationship between log-likelihood of DestiFlow and parameters $\alpha$ and $\beta$. (**a**) The relationship between log-likelihood of *DestiFlow* and parameters $\alpha$ and $\beta$ on weekday. (**b**) The relationship between log-likelihood of DestiFlow and parameters $\alpha$ and $\beta$ on weekend.

### 4.2.2. Evaluation Method

In the context of network theory, a complex network is a network with non-trivial topological features, the features often occur in networks modelling of real systems. The complex network is often used to analyze the properties and relationships between entities in the real world. The spatio-temporal flow model establishes the connection between the aggregation areas in time and space through the flow of dockless shared bikes. Therefore, the flow can be described by a network, which is not only used by many studies [3,36,37], but also provides a perspective to view the flow characteristics of dockless shared bikes.

A dynamic directed network $G = (V, E(t), W)$ is used to describe the spatio-temporal flow of dockless shared bikes. $V$ is a set of nodes in the network and represents the aggregation areas of dockless shared bikes. $E(t)$ is a set of edges in the network, which is the directed edge from the departure aggregation area to the arrival aggregation area. The connection of network changes with time. $W$ is a set of weights, indicating the number of edges between departure aggregation area and arrival aggregation area.

In this paper, we evaluate the experimental results by using the topological features of the spatio-temporal flow network of the dockless shared bikes. Due to the irregular flow of dockless shared bikes, we concern more about whether the model can describe the topological features of the flow network. The traffic and hot spots of dockless shared bike flow can be accurately grasped through the topological features. Three metrics of network centrality, namely degree centrality, betweenness centrality, and closeness centrality [38,39], are used to evaluate the experimental results:

**Degree centrality**: The degree centrality for a node $u$ is the fraction of nodes connected to it:

$$D(u) = \frac{k(u)}{N-1}. \tag{26}$$

where $k(u)$ represents the degree of node $u$, $N$ represents the number of nodes. The higher degree centrality of a node indicates that the node plays a more important role in connecting other nodes.

**Betweenness centrality**: The betweenness centrality of a node $u$ is the sum of the fraction of all-pairs shortest paths that pass through $u$:

$$B(u) = \sum_{s,t \in V} \frac{\sigma(s,t|u)}{\sigma(s,t)}. \tag{27}$$

where $\sigma(s,t)$ is the number of shortest (s, t)-paths, and $\sigma(s,t|u)$ is the number of those paths passing through node $v$ other than $s$, $t$. The higher the betweenness centrality of a node, the more obvious the role of the node as a bridge connecting other nodes.

**Closeness centrality**: The closeness centrality of a node $u$ is the reciprocal of the average shortest path distance to $u$ over all $n-1$ reachable nodes:

$$C(u) = \frac{n-1}{\sum_{v=1}^{n-1} d(v,u)}. \tag{28}$$

where $d(v,u)$ is the shortest-path distance between $v$ and $u$, and $n$ is the number of nodes that can reach $u$. The higher closeness centrality of a node indicates that the node is closer to other nodes.

These three centrality metrics show the importance of nodes in the network from different perspectives.

### 4.2.3. Results

The flow network of dockless shared bike extracted from the real data is called the real network, and the flow network obtained from the experiment is called the generated network. Figure 7 shows the comparison of the centrality distributions between the real network and the generated network. In the figure, the degree centrality distribution, the betweenness centrality distribution and the closeness centrality distribution of the real network and the generated network are compared. It can be found that the tail of degree centrality distribution and betweenness centrality distribution follow the power-law distribution, indicating that the degree centrality and betweenness centrality of most nodes are very small and the importance of these nodes is also very small, only a small number of nodes are important. There are two parts of the closeness centrality distribution, only a few nodes have a very small closeness centrality, and most nodes have a very high closeness centrality. This indicates that the average shortest-path distance is short, which conforms to the characteristics of dockless shared bikes as a short-distance transportation. In addition, we compare the three centrality metrics of generated network at different times. Figure 8 shows the comparison of the centrality distributions between the generated network in 5 December 2017 and the generated network in 22 May 2018. In the figure, there are more nodes with high degree centrality and closeness centrality in summer, indicating that the aggregation areas with high activity will be more, and the connections between these aggregation areas will be closer in the summer. In general, the centrality distributions of the real network and the generated network is consistent. The spatio-temporal flow model can well reflect the importance of aggregation areas and describe the structural characteristics of the flow network.

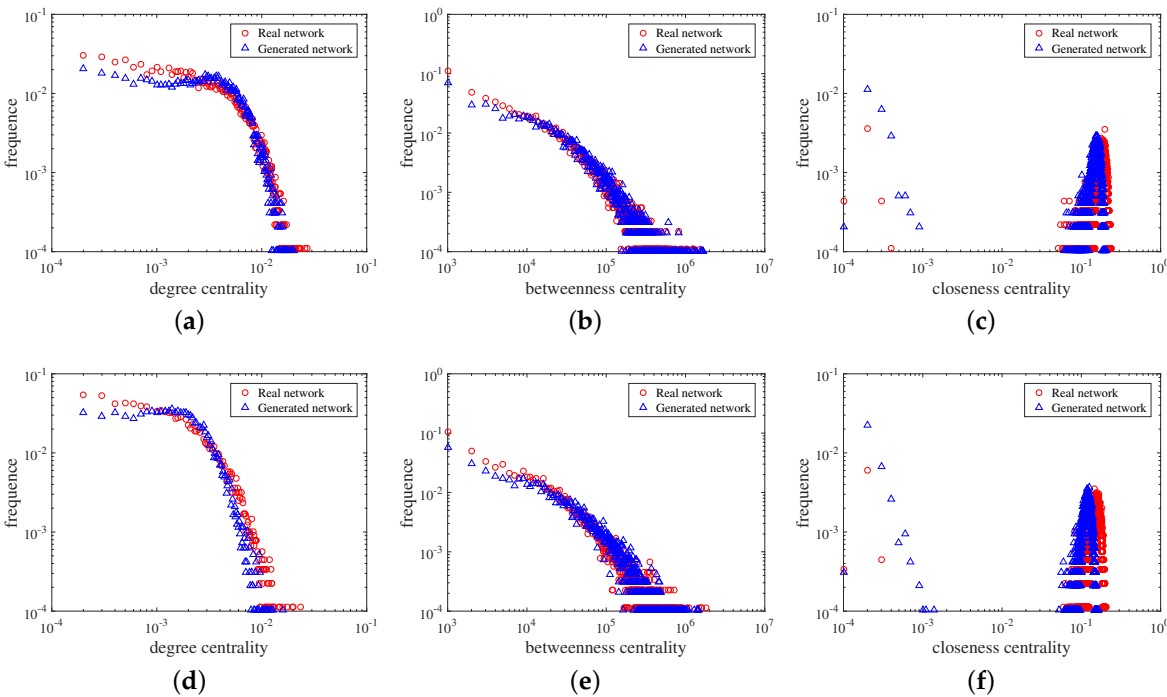

**Figure 7.** The comparison of the centrality distributions between the real network and the generated network. (**a**) Degree centrality distribution on weekdays. (**b**) Betweenness centrality distribution on weekdays. (**c**) Closeness centrality distribution on weekdays. (**d**) Degree centrality distribution on weekends. (**e**) Betweenness centrality distribution on weekends. (**f**) Closeness centrality distribution on weekends. The red circle represents the real network, and the blue triangle represents the generated network.

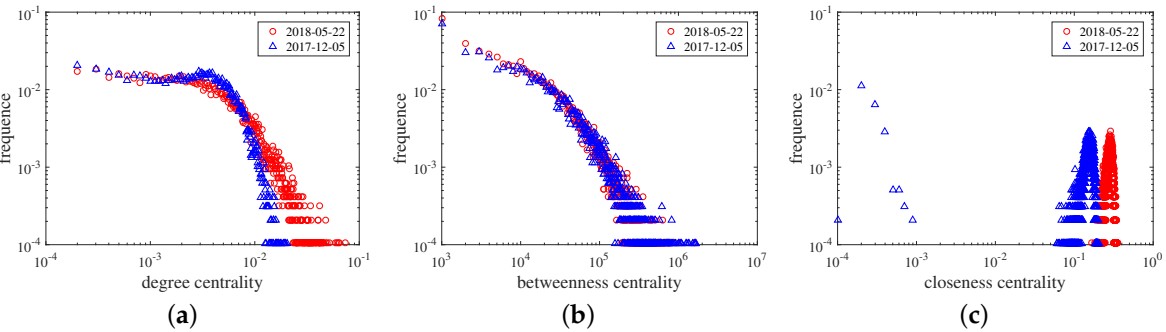

**Figure 8.** The comparison of the centrality distributions between the generated network in 5 December 2017 and the generated network in 22 May 2018. (**a**) Degree centrality distribution. (**b**) Betweenness centrality distribution. (**c**) Closeness centrality distribution. The red circle represents the real network, and the blue triangle represents the generated network.

## 5. Case Analysis

The *DestiFlow* model studies the flow of dockless shared bikes from the perspective of aggregation areas. The model can be used to predict the demand for dockless shared bikes in each aggregation area and help solve the problem of unbalanced spatial distribution of dockless shared bikes. For this purpose, a case study is carried out. Using the number of dockless shared bikes in each aggregation area at 00:00 on 6 December 2017, the real-time number of dockless shared bikes in each aggregation area can be predicted. Figure 9 shows the results of the case study. For better presentation of the results, the results of a subspace (longitude range [116.310, 116.345], latitude range [39.910, 39.930]) are shown. Figure 9a shows the change of the predicted number of dockless shared bikes in each aggregation area over time. Different color lines represent different aggregation areas. By predicting the change of the

number of dockless shared bikes in each aggregation area, the demand for dockless shared bikes can be effectively grasped. At a certain time, the number of dockless shared bikes in each aggregation area is different, with a maximum of over 200 and a minimum of 0. The difference in the number in different aggregation areas reflects the unbalance spatial distribution of dockless shared bikes. Figure 9b shows the number of dockless shared bikes in each aggregation area at 00:00 on 6 December 2017 in geographical space, and Figure 9c shows the change of the number of dockless shared bikes in each aggregation area on 6 December 2017. The change of the number of dockless shared bikes in each aggregation area within one day is defined as the difference between the number of dockless shared bikes in a gathering area at 00:00 and the number of dockless shared bikes in the gathering area at 24:00. Three problems can be found from Figure 9: (1) some aggregation areas have a high number of dockless shared bikes, but few bikes are used, which will cause a waste of bikes, such as aggregation area 1; (2) the arrivals of dockless shared bikes in some aggregation areas is much higher than the departures, which will lead to the accumulation of dockless shared bikes for a long time, such as aggregation area 2; (3) the number of dockless shared bikes in some aggregation areas is not very large, but the usage is very high, which will cause the shortage of dockless shared bikes, such as aggregation area 3. The unbalanced spatial distribution of dockless shared bikes is not conducive to the reasonable utilization of resources, so it is necessary to carry out a reasonable scheduling. Our model can predict the short-term flow of dockless shared bikes in the future through historical data, so as to predict the demand for dockless shared bikes, and guide the scheduling in advance.

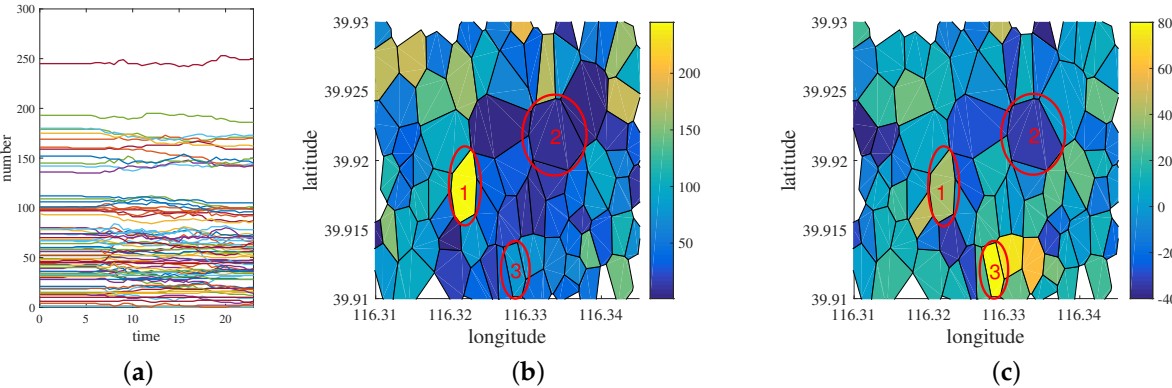

**Figure 9.** The results of case analysis. (**a**) The change of the predicted number of dockless shared bikes in each aggregation area over time. Different color lines represent different aggregation areas. (**b**) The number of dockless shared bikes in each aggregation area at 00:00 on 6 December 2017 in geographical space. The value corresponding to the color is shown in the color bar. (**c**) The change of the number of dockless shared bikes in each aggregation area within one day.

## 6. Discussion and Conclusions

In this paper, the flow characteristics of dockless shared bikes are expounded through the analysis of the time series location data of ofo and mobike shared bikes in Beijing. Based on the analysis, a model called *DestiFlow* is proposed to describe the spatio-temporal flow of urban dockless shared bikes.

Unlike previous work, which studies the crowd flow of shared bike from a macro perspective [13], the *DestiFlow* method can describe in detail the flow of each dockless shared bike. In addition, the POI clustering provides a new method for discovering the aggregation areas of dockless shared bikes, which avoids the shortcomings of the method using a regular geographic grid [14,15].

It is found that the *DestiFlow* method can reflect the importance of aggregation areas and describe the structural characteristics of the flow network. Through the case study, it is found that the spatial distribution of dockless shared bikes is unbalanced. Our model can guide the scheduling in advance by predicting the demand for dockless shared bikes in each aggregation area. The results can help to grasp the mobility characteristics of citizens and improve the urban traffic management system. The flow of dockless shared bikes is dominated by human mobility behavior. An in-depth understanding of the

flow characteristics of dockless shared bikes can help us master the short-distance human movement pattern, which is beneficial to the construction of intelligent transportation and intelligent city.

However, there are still some limitations in our work. (1) The usage mode of dockless shared bikes needs to be further explored. This paper analyzes the characteristics of flow distance and time of dockless shared bikes, but these are considered from a macro perspective. More detailed studies can be considered, such as the flow characteristics of dockless shared bike in different locations. (2) The model needs to be further improved. The purpose of our model is to study the flow of dockless shared bikes in the short term. If multivariate data can be integrated and the influence of weather, temperature, population, economy and other factors can be fully considered, the accuracy and practicability of the model can be improved.

**Author Contributions:** Conceptualization, Jian Dong and Chuan Ai; methodology, Jian Dong; software, Jian Dong; validation, Jian Dong, Bin Chen, Lingnan He and Chuan Ai; formal analysis, Jian Dong, Fang Zhang; investigation, Fang Zhang; resources, Bin Chen, Danhuai Guo, Lingnan He, Xiaogang Qiu; data curation, Danhuai Guo, Fang Zhang; writing–original draft preparation, Jian Dong; writing–review and editing, Jian Dong, Bin Chen, Chuan Ai; visualization, Jian Dong; supervision, Jian Dong, Bin Chen, Lingnan He, Danhuai Guo and Xiaogang Qiu; project administration, Jian Dong; funding acquisition, Bin Chen, Lingnan He.

**Acknowledgments:** This study is supported by National Key Research & Development (R & D) Plan under Grant No. 2018YFC0806900 and the National Natural Science Foundation of China under Grant Nos. 71673292, 71673294 and National Social Science Foundation of China under Grant No. 17CGL047 and Beijing National Science Foundation of China under Grant No.91224006 and Guangdong Key Laboratory for Big Data Analysis and Simulation of Public Opinion.

**Conflicts of Interest:** The authors declare no conflict of interest.The funders had no role in the design of the study; in the collection, analyses, or interpretation of data; in the writing of the manuscript, or in the decision to publish the results.

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
