# Peer review of "A Spatio-Temporal Flow Model of Urban Dockless Shared Bikes Based on Points of Interest Clustering"

_ijgi, doi:10.3390/ijgi8080345_

Round 1
Reviewer 1 Report
The article entitled "A Spatio-temporal Flow Model of Urban Dockless Shared Bikes Based on Point-of-Interests Clustering" introduces algorithms for aggregating locations of freely parkable public city bikes and for describing and forecasting their locations in order to enhance the spatial balancing of bikes and to better understand movements of citizens. The introduced method is novel in using point-based clustering instead of coverage-based data and the clustering is innovative in its gathering around POIs. The article is logically consistent and certainly interesting to the readers of the IJGI and can guide practical implementations of bike-sharing. However, there are certain weaknesses in the article that should be improved on its way to publication, which is why I suggest a major revision although I find the article of high value.
At first, some general notions to be improved on. The literature review in the article is purposeful but unnecessarily concise. At least, you should elaborate some central notions from docked shared bikes (L26-27), present known subjective factors of bike demand (L40-41) and tell about specific flow behaviours of shared bikes (L48).
Also, description of the origins and characteristics of data is not satisfying (Subsection 2.1). I think it is not you who collected the dataset as you write, so please specify where you got the data from. Numerical descriptors of location data are also necessary, particularly how the location data are recorded? Do you know some routes and some not, why (L92)?
In addition, the paper misses proper discussion that is a central aspect of every scientific study. You discuss the results well while presenting them but mention only very shortly a couple of limitations in your work (L365-371) and do not give overall description of your outcome in relation to others'. Please describe these more profoundly, relate your results to the previous literature and give a paragraph on what in the end is your original contribution to the topic and geoinformatics.
Second, a couple of figures and their interpretations require clarification in order to serve the reader. As of Figure 2, you conclude that the flow distance distribution of the shared bikes is time stable. However, according to the figure the JS divergence varies much across time, so please elaborate why do you state the stability. Also, why do you use JS divergence and not other distribution comparison statistics?
As of Figure 3, you conclude that citizens would be more inclined to use shared bikes in the summer because the peaks in the figure are higher. To my view this is a false statement from departure rates. You should show absolute amounts to state about the use amounts of bikes, and find another explanation for the differences in temporal rate distributions of Figure 3.
Third, the language of the article is generally good but I found many either linguistically or logically unclear and imperfect wordings which must be enhanced to make the article well readable (a careful proof-reading would also remove more typos or minor linguistic mistakes):
- "point-of-interests" -> "points of interest"
- L3 "related research": too vague, what kind of research?
- L9 "rebalance": I think your model does not rebalance the bike distribution but shows the imbalance
- "docking shared bikes": sounds grammarly inconvenient term, better e.g. "dock shared bikes"
- L28 "few" -> "only few"
- L36-37: "method of regular geographic grid" -> "method using..."
- L54-55: "from its specific location": unclear wording, please rewrite
- L71,72 (elsewhere?): "Denstiflow" -> "DensiFlow"
- L94 "in Manhattan": the region is not Manhattan, rewrite
- L109 "is introduced": you're not introducing but describing something already made
- L111 and further, "probability distribution": not only probability but also observed distributions
- Figure 1 caption, "red circle": to me the symbols look like triangles
- L183 and elsewhere, "category": what you mean by category remains unclear throughout the text - typically it would be classes of POIs but it seems to me this is not what you mean
- L185-186: you should refer to a figure with clustering regions
- L223 "scales of cluster": please be more exact
- L242 "obviously superior": strong words. Why so? You must elaborate. The consequent suggestion is also unargumented.
- Figure 4: Colors of clusters in 4a are not separable by eye - e.g. randomise their order like in 4b. In 4c-e, the title of the horizontal axis must be more exact - what number?
- L254 "better to divide" etc.: unclear and grammatically inconsistent sentence, please rewrite and elaborate.
- L281 "The complex network": please elaborate
- L338 "significant differences": the differences in the figures do not seem large to me, please use relative measures.
- Figure 7: The caption is unnecessarily verbose.
Author Response
Thanks for your comments on our manuscript entitled “A Spatio-temporal Flow Model of Urban Dockless Shared Bikes Based on Point-of-Interests Clustering” The manuscript has certainly benefited from these insightful constructive revision suggestions. We have carefully studied the comments as well as suggestions, and have made corresponding corrections. Revised contents are marked in yellow in the manuscript as well as this response letter. The points mentioned by the reviewer are discussed as follows.

Reviewer 2 Report
The authors propose to apply POI-based clustering techniques in order to analyze the spatial distribution of dockless shared bikes.
The assumptions made to model the spatial and temporal distribution of the bikes are not well specified. Please provider deeper insight into the theoretical models behind the performed analysis. For example, the use of a Gaussian mixture model seems to be a little bit arbitrary.
The evaluation part is quite detailed. However, the considered data seems to be quite limited. Please provide more statistics on the analyzed data and empirically compare the results achieved in different periods of time.
Author Response

(The authors gave the same response as above.)

Round 2
Reviewer 1 Report
The authors have made appropriate effort in revising the article and I am ready to propose its publication. However, I still see a few issues that must be corrected before publication:
- Now that the subfigures in Figure 2 illustrate different phenomena, the caption requires revision - it is not only JS divergence anymore. For the subfigure 2c, the caption should name the time from 2b.
- The time unit in Figure 3 must be named. Also, how can Figure 3 be exactly the same as in the previous version although you write that you have changed from the relative rate to the absolute number - the total number of departures per day cannot be the same all the time? Even if this was the case, your conclusion seems still wrong - e.g., why would bikers ride less during the day in the summer than in the winter? In addition, the caption has not been changed according to the figure, which must be done.
- As of Figure 9, if you state that there are differences, you must make those differences visible or explain them better. I do not see but very minor changes in the numbers of bikes according to the curves - mostly the numbers keep quite the same. Also, what day are you showing in 9c and why just that day?
- L340-1 "high activity in summer will increase": Clarify if you mean that there are more areas of high activity or activity is higher in the areas. The consequent sentence is also grammatically incorrect.
Author Response
Thanks for your comments on our manuscript entitled “A Spatio-temporal Flow Model of Urban Dockless Shared Bikes Based on Points of Interest Clustering”. The manuscript has certainly benefited from these insightful constructive revision suggestions. We have carefully studied the comments as well as suggestions, and have made corresponding corrections. Revised contents are marked in yellow in the manuscript as well as this response letter. The points mentioned by the reviewer are discussed as follows.

Reviewer 2 Report
The authors have addressed the reviewers' concerns.
Author Response
Thanks for your comments on our manuscript entitled “A Spatio-temporal Flow Model of Urban Dockless Shared Bikes Based on Points of Interest Clustering”. We appreciate for editor and reviewer’s excellent work earnestly, and we hope that the corrections will meet with approval. Thanks for your help with our paper processing again.

This manuscript is a resubmission of an earlier submission. The following is a list of the peer review reports and author responses from that submission.
Round 1
Reviewer 1 Report
This paper proposes the Destiflow method to model the flow of dockless shared bikes. The main idea of this paper is to first generate the aggregation areas by clustering the locations of bikes and POIs and then model the bike flow based on spatio-temporal probability methods. It is interesting and useful to capture the flow characteristics of the dockless shared bike system. However, I have some comments and suggestions listed as follows:
1. According to the experimental sections, the final goal of this paper is to model the flow network of dockless shared bikes in one day. Although it is easy follow for each stage including aggregation area clustering and flow modelling by Gaussian distribution models, the connection between these two stages is not clear. For example, what role does the type of aggregation area play? In addition, the definition of aggregation area (Eq. 3) is not clear. Because the aggregation area is a circle range, what are the longitude and latitude of an aggregation area?
2. The experimental data only cover 6 days in December. I think that the scalability is too small. I am interested in whether the analysis results are similar in spring, summer or fall. Hence, I think that more experiments are necessary.
3. The parameter testing in Section 5.3 is arbitrary. In Figure 7, this paper only test four parameter combinations and select the best one according to the minimum mean square error. Parameter setting requires a more rigorous approach.
4. There are five topological features used to evaluate the experimental results. I suggest that the authors can provide an example to explain them. I think that the number of nodes or edges is easy to understand. However, it is not easy to understand how to calculate the density of a flow network.
Some minor issues:
1. In most of papers, POI is point-of-interest instead of interest-of-point.
2. There are some duplicate sentences between Figure 2 and lines 123-126 of p.5.
3. There is no description about why the Y function is Eq. 17.
4. In line 283, figure 8 should be corrected to figure 7.
Reviewer 2 Report
The authors of the paper describe a clustering method based on the interest of points with a goal to find the aggregation areas of dock-less shared bikes. The proposed method avoids the problems of the existing clustering methods based on geographic grid which makes it suitable for the analysis of dock-less shared bikes. Through the analysis of the time series location data of the shared bikes in Beijing, the flow characteristics of the dock-less shared bikes are analyzed in terms of distance and activity. A model called Destiflow is proposed to describe the spatio-temporal flow of the dock-less shared bikes.
There are several things that I liked about this paper. I appreciate that fact that the authors are using a real dataset which makes the experimental evaluation more valuable. I also liked the detailed review of the existing grid based methods for clustering of bikes.
The paper however has a lot of serious issues. First of all - I am not sure that I understand what is the problem that you are trying to solve. Having model for the sake of it is useless. This model needs to be used to answer real life questions for the specific domain. I did not see a single one of those questions.
Second - your proposed method for clustering is not that different from the partitioning method used by other authors. They might use grid - which is not necessary regular one. You use points of interest to seed the clustering which should lead to similar results. You should compare with one of the other methods to prove that you get better results.
Third - the paper has a lot of language issues. I would recommend to have a native speaker proof read it before resubmitting it again.
Overall - this paper needs a lot more work.